# CRISPR/Cas9 for Insect Pests Management: A Comprehensive Review of Advances and Applications

Sanchita Singh [1,2,†], Somnath Rahangdale [1,3,†], Shivali Pandita [1,4], Gauri Saxena [2], Santosh Kumar Upadhyay [5], Geetanjali Mishra [4] and Praveen C. Verma [1,3,*]

1. CSIR-National Botanical Research Institute, (Council of Scientific and Industrial Research) Rana Pratap Marg, Lucknow 226001, UP, India
2. Department of Botany, University of Lucknow, Lucknow 226007, UP, India
3. Academy of Scientific and Innovative Research (AcSIR), Ghaziabad 201002, UP, India
4. Department of Zoology, University of Lucknow, Lucknow 226007, UP, India
5. Department of Botany, Panjab University, Chandigarh 160014, UP, India
* Correspondence: praveencverma@nbri.res.in
† Both the authors contributed equally to this manuscript.

**Abstract:** Insect pests impose a serious threat to agricultural productivity. Initially, for pest management, several breeding approaches were applied which have now been gradually replaced by genome editing (GE) strategies as they are more efficient and less laborious. CRISPR/Cas9 (Clustered Regularly Interspaced Short Palindromic Repeat/CRISPR-associated system) was discovered as an adaptive immune system of bacteria and with the scientific advancements, it has been improvised into a revolutionary genome editing technique. Due to its specificity and easy handling, CRISPR/Cas9-based genome editing has been applied to a wide range of organisms for various research purposes. For pest control, diverse approaches have been applied utilizing CRISPR/Cas9-like systems, thereby making the pests susceptible to various insecticides, compromising the reproductive fitness of the pest, hindering the metamorphosis of the pest, and there have been many other benefits. This article reviews the efficiency of CRISPR/Cas9 and proposes potential research ideas for CRISPR/Cas9-based integrated pest management. CRISPR/Cas9 technology has been successfully applied to several insect pest species. However, there is no review available which thoroughly summarizes the application of the technique in insect genome editing for pest control. Further, authors have highlighted the advancements in CRISPR/Cas9 research and have discussed its future possibilities in pest management.

**Keywords:** CRISPR/Cas9; insect pest; integrated pest management; genome editing

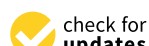



## 1. Introduction

Annually, phytophagous insects damage one fifth of the world's total crop yield. Biotic stress affects the food security of any country by compromising the quality and quantity of the crop productivity. The yield loss due to insect infestation has devastating impacts on the society such as, hunger and poverty. The combined impact of the emergence and/or re-emergence of insect pests and rapidly growing human population calls for immediate inventions and the use of rigorous and integrated agricultural practices. The FAO estimated that plant diseases and pests are responsible for a 20–40% reduction in the global crop yields per year [1]. Researchers predicted a strong decline in the crop yield in response to climate change and weather pattern variations. Climatic changes might increase the risk that is caused by phytophagous pests, thereby turning them into a more harmful threat to the crops [2]. Over the past thousands of years, plant breeding has been exploited for constructing insect resistant crop varieties, however, it is laborious, time-consuming, has a stochastic nature, and the screening process is a very challenging practice [3]. Further, the unavailability of a resistance source in the gene pool has restricted

the scope of breeding a resistant cultivar [4,5]. Under such a scenario, the use of toxic and cost-intensive agrochemicals appeared to be the only convenient solution for crop protection. Considering the toll these chemicals take on the ecosystem, there was an urge to develop genetically stable and fixed plant types [6].

Genome editing (GE) can play a pivotal role as it is a more promising and an environmentally friendly answer that can be used to deal with the situation. It all began with the gene-targeting experiments on the protoplast of *Nicotiana tabacum* which were performed in 1988 [7] and the findings in 1993 which supported that DNA double-strand breaks (DSBs) improved the gene-targeting efficacy [8]. Since then, the scientific orientation shifted towards the development of targeted genome editing techniques. The adoption of GE systems provided remarkable results in the field of the genetic improvement of crops. Genetic engineering rationalized the biological research world with the introduction of methods involving in vivo genome editing. The GE technique results in base substitutions and/or insertions/deletions (indels) in the target DNA. It includes several techniques, for instance, the use of zinc finger nucleases (ZFNs), transcriptional activator-like effector nucleases (TALENs), and the recently established clustered regularly interspaced short palindromic repeat (CRISPR)/CRISPR-associated nuclease 9 (Cas9) system. In contrast to TALENs and ZFNs, the CRISPR/Cas system is more direct and easier to handle as it requires a single guide RNA (gRNA) for target determination with the Cas9 nuclease [8,9]. In the recent past, the preference has shifted from breeding insect-resistant cultivars to making CRISPR/Cas9-mediated modifications in the agronomic traits or targeted mutagenesis in the insect genome. The constant modifications to gene knockout strategies, transgene integration, nucleotide substitution, transcription regulation, etc., have made the CRISPR/Cas system an easy-to-apply, cost-effective, and a widely used technique for manipulations at the genetic levels [10–12]. Biotic stress resistance is one of the traits that is improved by GE, which makes CRISPR/Cas system highly efficient in enhancing global food security, crop protection, and sustainable agriculture (Figure 1). On the subject of insects, many research groups have reviewed the application of various GE techniques, with special attention being paid to the CRISPR/Cas9 system in arthropods; in spite of this, no inclusive report is available which covers all of the insect pests. The recent developments in the field of molecular biology and omics approaches presented that there has been a peak in the usage of CRISPR/Cas9 technology for insect pest management, and the data from these reports are not summarized in any previously published reviews. In this study, the authors emphasize and explain the prospects and applications of CRISPR/Cas technology in different insect groups for pest management. The CRISPR/Cas9 system has the potential of providing promising approaches for the control of insect pests. Therefore, summing up the CRISPR/Cas9-based control strategies against insect pests is significant in achieving global goals such as sustainable development.

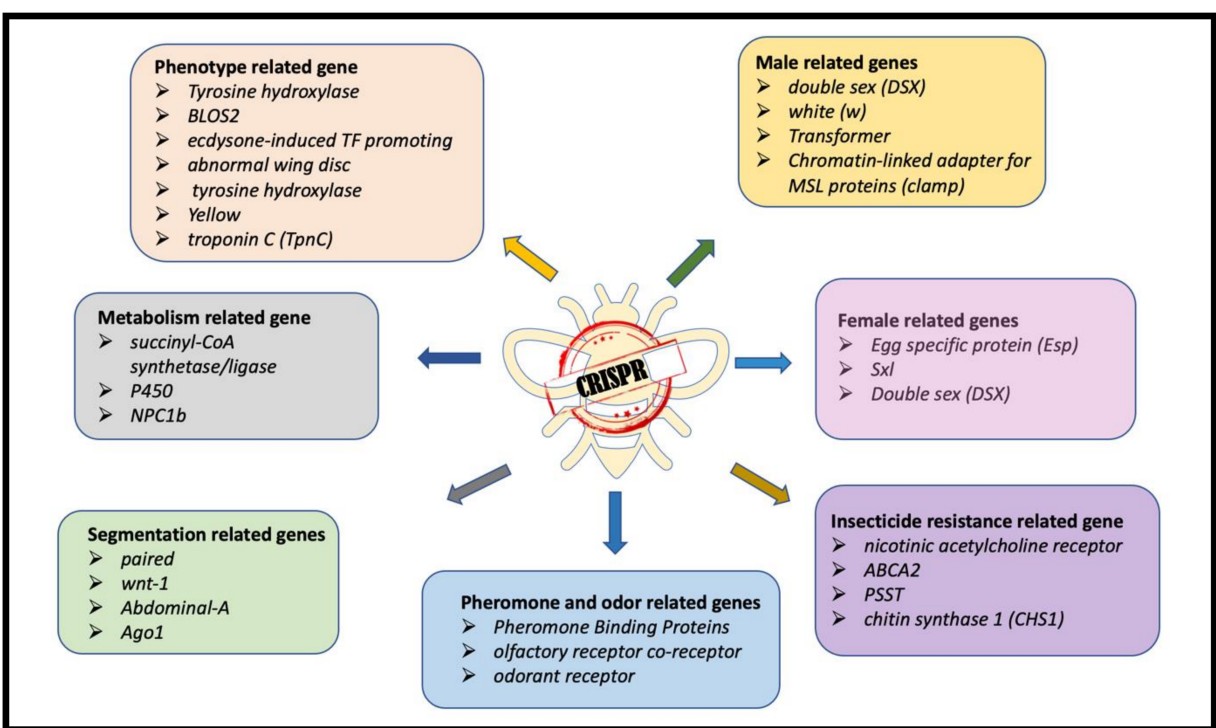

**Figure 1.** The figure shows various genes targeted by CRISPR/Cas9 for insect pests control.

### 1.1. CRISPR/Cas-Mediated Genome Editing in Insects

Biotechnology and molecular biology have experienced a great transformation and advancement since the development of the CRISPR/Cas9 gene-editing system in the mammalian cells in the year 2012 [13]. With the discovery of the homology-dependent cleavage recombination mechanism, the utility of CRISPR/Cas9 was explored in genome editing, and it emerged as an assuring genome editing tool. The growing utility of gene-editing tools as a site-specific genome-editing approach offers infinite opportunities which may have an impact on important agronomic traits such as resistance to biotic stresses [14–17]. The present study reviews the novel opportunities that CRISPR/Cas9 system offers and how it has attracted all of the attention by offering distinct advantages in the area of insect pest control thus ensuring a good crop yield and food security. A combination of gene editing tools and insect manipulation methods have been already used in *Drosophila melanogaster* Meigen, several tephritids, and mosquitoes which have answered some basic questions about insect biology. Recently, the technology has been utilized for the development of novel pest control strategies [18–21], and it has been proven to be an efficient approach for pest management [22–24]. The advancements in genome editing methods paved the way for inventive pest control methods by the development of genetically modified insects. The CRISPR/Cas technology is evolving as it is very beneficial for efficient tailoring and gene manipulation. The components of the CRISPR/Cas tool (sgRNA and the Cas9 protein) can be delivered in the target organism in form of plasmid DNA, RNA or a ribonucleo-protein (RNP) complex [25]. Some of the phytophagous insect orders that have been explored for pest management using genome editing by the CRISPR/Cas system are reviewed in this article (Table 1).

### 1.2. Diptera

Diptera is one of the largest insect orders. The insects of this order are two-winged and are called true flies. Dipterans are abundantly present all around the world, thus, the order is diversified. Their larvae are serious agricultural pests on solanceae, cucurbitaceae and other crop plant families.

### 1.2.1. *Drosophila*

As it is the model organism, *D. melanogaster* encouraged and improved the genome editing technology in insects through use of CRISPR/Cas9 [25]. Gratz et al. 2013 were the first to report the CRISPR/Cas9 technology-based deletion of 4.6 kb of chromosomal DNA *Drosophila* genome. The deletion was performed in yellow locus by using two target sgRNAs and an ssODN (single-stranded oligonucleotide donor) template [26]. Further, the possible applications and benefits of CRISPR/Cas9 in the designer flies generation were discussed [27]. In 2013, scientists developed a strategy to increase the homologous recombination (HR) frequency by utilizing a reintegration vector [28], equated the efficacies of CRISPR/Cas9-mediated and TALEN-based homology-directed repair (HDR), made an easily screen-able platform, and established three different HDR methods for site-specific mutagenesis [29].

Two different types of parent flies were developed; one was developed with the *Cas9* gene under the germline specific promoter, and another was developed, expressing sgRNA in a constitutive manner. The crossing of these parents produced progenies with a transmitted mutation in the germlines. In a similar way, a gRNA-encoding DNA vector was injected into the Cas9-positive transgenic flies which permitted the knockin or knockout of various genes targets [30–36]. Different approaches have been tried to achieve the induction of HDR in flies for example, a gRNA plasmid and donor repair template were injected into transgenic Cas9 embryos [37], while the transgenic embryos containing a sgRNA and Cas9 were injected with a donor template plasmid. All of these three components containing the *Cas9* gene, sgRNA and donor repair template in a plasmid form were delivered into non-transgenic flies [38,39].

A comparison was then made among the different approaches of facilitating HDR utilizing the same donor plasmid and gRNA, and it was observed that the use of non-transgenic individuals produced lower frequencies of the knockin ones when they were compared to the transgenic ones [40]. A mutagenic chain reaction (MCR) method was developed by Gantz et al. in 2015 that could lead to autocatalytic mutations, and they utilized this process for the conversion of mutations from heterozygous to homozygous forms [41]. The CRISPR/Cas9 technique was applied for the introduction of a site-specific mutation in the Dα6 subunit of the nicotinic acetylcholine receptor (nAChR). The resistance level for Spinosad-based insecticide for the flies with a Dα6-null mutation was higher than that of the site-specific mutation, and it was demonstrated that the site-specific mutation is directly related to spinosad resistance [42]. Studies have also focused on the altered phenotypes, which were obtained after the knockout of specific genes using the CRISPR/Cas9 system [43].

To show the direct relation between the efficiency of the CRISPR/Cas9 system and the concentration of sgRNA, the *yellow* gene was studied, however, the use of increased sgRNA concentrations reduced the adult survival rate [44]. The *yellow* gene was also targeted by Yu et al. 2013, and a great increase in the editing efficiency was demonstrated [45]. Through the CRISPR/Cas9 system, researchers introduced site-specific mutations into the *white* (w) gene of the *D. melanogaster* and *sex-lethal* (Sxl) genes of *D. suzukii*. The phenotype which was obtained by the mutation in the eyes was produced at a low efficiency, and this might have happened because the sgRNA and Cas9-encoding plasmids DNA were injected into the flies, and here, the plasmid DNA needs to be transcribed in vivo, which is not the case with mRNA. Additionally, the specificity of the *Drosophila* species and the white gene may have led to this low efficiency. Mutant females of the *Sxl* gene demonstrated abnormal genitalia and reproductive tissues [46]. The CRISPR/Cas9-mediated mutation in the alpha subunit of succinyl-CoA synthetase/ligase (Scsα) in *Drosophila* revealed that Scsα deficiency displayed developmental delays, increased mortality under starvation conditions and impaired locomotor activity [47]. Hence, for energy metabolism in *Drosophila*, the *Scsα* gene is vital. Asaoka et al. used CRISPR/Cas9 technology to create flies lacking a linear ubiquitin E3 ligase (LUBEL). When these LUBEL-deficient flies were exposed to heat, defective climbing and a reduced survival rate was observed [48].

Further, the utilization of the CRISPR/Cas9 system for vital gene mutation such as the *clamp* gene, *troponin C* (*TpnC*), *Alk* gene, Sex-lethal (Sxl) and white (w) gene in *Drosophila* has laid a strong foundation for the development of sustainable pest management strategies [49–53]. The sterile insect technique (SIT) is a pest management approach that includes rearing, sterilizing and releasing the sterilized males of the target insect species. The sterile insect technique is an environmentally friendly and successful strategy for pest control. It is a tool that can be used for elementary research on the reproductive biology of pests. The integration of SIT with CRISPR–Cas genome editing might create optimal SIT strains [54,55]. The site-specific editing of the white gene by directly delivering the purified Cas9 protein in the embryo of the *D. suzukii* simplified the gene editing. For the generation of heritable genetic modifications, a recombinant Cas9 protein could be a way of choice. Choo et al. in 2017 generated a CRISPR/Cas-mediated series of frameshift mutations, leading to genetic sexing strains in *Bactrocera tryoni* (Froggatt). The knowledge from this study could be exploited for SIT-based pest control [56]. In context to SIT, the system of sperm marking helps to monitor the pest population. In *D. suzukii*, a sperm-marking transgenic strain was developed by using endogenous promoters of *D. suzukii* in 2019 [57]. The Ds hsp70 promoter is used to derive the expression of Cas9 and a small nuclear RNA gene U6 promoter for the expression of gRNA. In this particular study, the co-injections of the helper plasmid were found to be more effective over the preformed RNPs that had been used previously in HRD-based GE [58]. The CRISPR/Cas9-mutated white (w-) gene in *D. suzukii* caused a copulation failure. It also caused a pigmentation deficiency in the testis sheath, which could be a probable reason for the copulation failure. This approach may, therefore, be capable of being used in pest management [57,59] (Figure 2A).

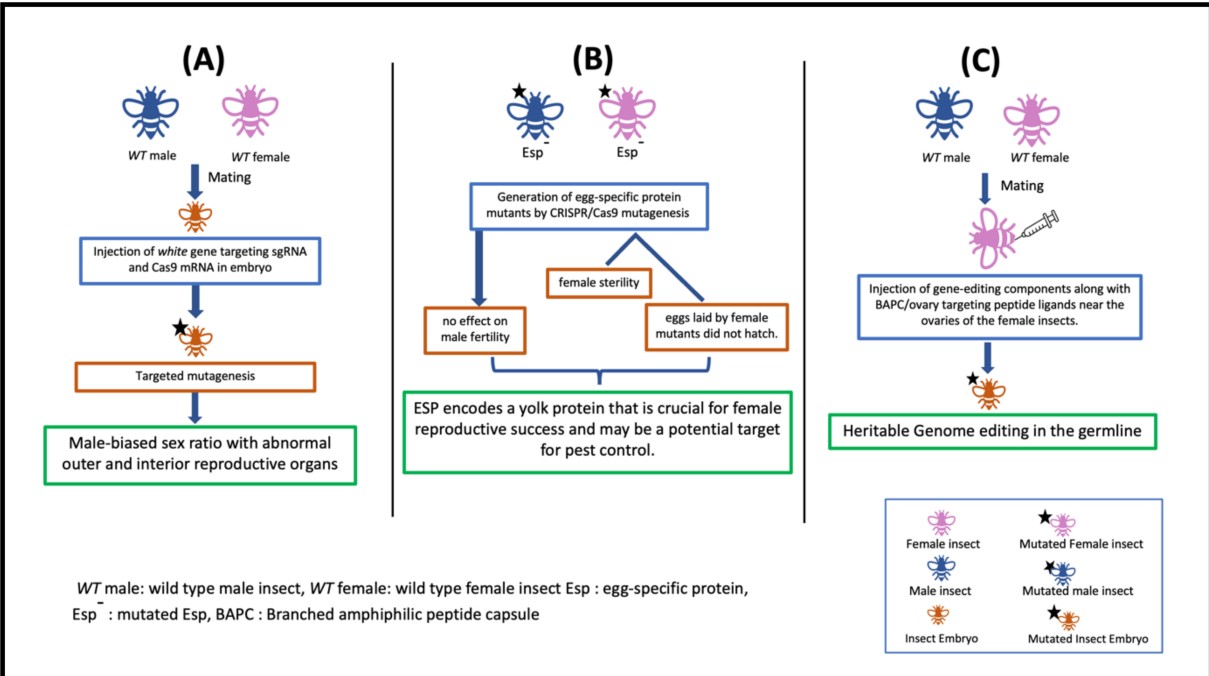

**Figure 2.** Figure shows three different CRISPR-based genome-editing approaches for insect pests management. (**A**) pictorial representation of sterile insect technique (SIT); (**B**) targeting female reproductive fitness through CRISPR/Cas9; (**C**) injection of CRISPR/Cas9 components with ovary targeting signals for producing heritable genome editing in the germline.

The Dipteran group, Tephritid, comprise destructive pests that are a threat to agricultural crops. Genetic approaches for pest management are very promising methods for tephritid population control. This involves genetic manipulations which enables the insect

to transmit lethal traits or renders them sterile. Some examples of genetic pest management in tephritidae are mentioned below.

### 1.2.2. *Anastrepha ludens*

*Anastrepha ludens,* which are also known as the *Mexfl* (North American Plant Protection Organization's Phytosanitary Alert System) belongs to the family of fruit flies. *A. ludens* is native to Central America and Mexico and is a chief pest of mango and agriculture in lower Rio Grande Valley, Mexico and Central America [60,61]. The species has a relatively long lifespan in comparison to other fruit fly species [62]. This quality is responsible for the aggressive invasion of *Anastrepha* spp. This genus poses a major threat to the yield of different species of fruit [63]. The *Anastrepha* genus is designated as one of three genera that cause a major risk to American agriculture [64]. As per the USDA, the Mexfly can cause an estimated damage of worth $1.44 billion in a 5-year time span. Recently, in 2019, Li. et al. used the CRISPR/Cas9 system for targeting the sex determination gene, *As-transformer-2* (*Astra-2*), through the embryonic injection of the RNP complex. They demonstrated that knocking out of this gene led to sterility in some males and intersexual phenotypes in XX chromosome females. Thus, exploring this sex determining gene (*Astra-2*) can be useful in pest control management [63].

### 1.2.3. *Bactrocera dorsalis*

*Bactrocera dorsalis* is a destructive pest that is found in Asian countries. The *white* and *transformer* genes were mutated using the CRISPR/Cas9 system. The sgRNA targeted to the *transformer* gene, and the Cas9 mRNA were co-injected in the *B. dorsalis* embryos. The mutations in the *transformer* gene led to a male-biased sex ratio and as well as this, abnormal outer and interior reproductive organs were formed. These mutations were heritable in the next generation, and thus, this gene can be a suitable target for controlling this pest [65].

### 1.2.4. *Ceratitis capitata*

The *Ceratitis capitata,* which is also called as the Mediterranean fruit fly, is an aggressive polyphagous pest that is responsible for a huge amount of economic damage. Initially, the eGFP-to-BFP conversion approach demonstrated that the CRISPR–Cas HDR genome editing technique utilizing short and ssDNA repair templates was highly efficient in *C. capitata*. Possibly, this efficiency could save resources and time during mutagenic screening while targeting the gene which does not show any phenotypic alterations [66]. Meccariello et al. in 2017 targeted the segmentation *paired* gene (*Ccprd*) and the eye pigmentation gene *white eye* (*we*) in *C. capitate* using the CRISPR–Cas9 system [67].

### 1.3. Lepidoptera

Lepidoptera is one of the world's most described species and comprises numerous taxa. Lepidopterans are the second most invasive pests of stored crop products, and thus, they have scientific and economic importance. Despite there being great interest in this group, there is unsatisfactory progress in the field of the genetic manipulation of them. Lepidopterans have devastating effects on crop yields. The overuse of chemicals to control the major Lepidopteran insect pests like *Spodoptera litura*, *Spodoptera littoralis*, *Helicoverpa armigera* and *Plutella xylostella* led to the development of resistance in the insects against the traditionally used pesticides. The two key reasons which explain the failure of the routine use of manipulative genetics in Lepidoptera were resistance for RNAi and sensitivity towards inbreeding [68,69]. Luckily, CRISPR/Cas9-mediated GE provided a heritable and ecofriendly solution for pest management. The GE approaches in some of the insect pests belonging to this order are reviewed in this report [70].

### 1.3.1. *Helicoverpa armigera*

Insects depend on food for their sterol requirements as they are unable to synthesize it. NPC1b is an insect protein that has been identified as an integral membrane protein.

The role of NPC1b in *Drosophila melanogaster* is an uptake of dietary cholesterol [71]. The CRISPR/Cas9-edited *NPC1b* mutant larvae were used to characterize the function of NPC1b in *Helicoverpa armigera. NPC1b* is vital for the growth and dietary cholesterol uptake of *H. armigera*, therefore, a limitation in the dietary uptake of cholesterol inhibits the weight gain and food ingestion of the insect. Thus, *NPC1b* can be a potential target for pest management. However, the technique might have some off-target effects [72]. Wang et al. used the CRISPR/Cas9 system in *H. armigera* and proved that Cry1Ac resistance relies on *HaCad* (which is also a key receptor of Cry1Ac). Instead of injecting plasmids coding the Cas9 and sgRNA, the eggs were injected with a mixture of Cas9 mRNA and sgRNA, and this resulted in a highly efficient editing of the *HaCad* locus. This study led towards the use of Bt toxins [73]. The CRISPR/Cas9 system also provided an innovative pest control approach against *H. armigera* by using the antagonist-mediated optimization of the time of mating which ensured maximum fecundity [74].

In *H. punctigera* and *H. armigera,* the Bt toxin Cry2Ab resistance was linked with a loss-of-function mutation in the ABC transporter gene (ABCA2). To confirm this correlation between the Cry2Ab resistance in *H. armigera and* ABCA2 gene, *HaABCA2* knockout strains were developed using the CRISPR/Cas9 technique. The knockout strain obtained a significant level of resistance, and this confirmed the role of HaABCA2 *in* intermediating the toxicity of Cry2Ab and Cry2Aa against *H. armigera* [75]. In *H. armigera,* four pigment genes, namely, *white*, *ok*, *brown* and *scarlet,* were also mutated using the CRISPR/Cas9 system which further led to several physiologically altered phenotypes [76]. For a better understanding of insect-insecticide response and the management of agronomic pests, the CRISPR/Cas9-mediated knock-out lines were made for the cluster of nine P450 genes in *H. armigera*. The study provided the basic information for the critical evaluation and identification of the main players in insecticide metabolism [77].

### 1.3.2. *Plutella xylostella*, *Spodoptera*, *Dendrolimus punctatus* and *Cydia pomonella*

The extra cellular matrix (ECM) of an insect is made up of chitin and protein. Chitin is a naturally occurring amino-polysaccharide that is synthesized by the *Chitin synthase 1 (CHS1)* gene-encoded protein. Some common examples of chemicals that directly hinder chitin synthesis are Benzoylureas (BPUs), etoxazole and buprofezin. In the *CHS1* gene, SNP was reported at position 1017, in which phenylalanine (F) replaced isoleucine (I), and this change was related to etoxazole resistance [78]. Another study on the mutation in the *CHS1* gene of *Plutella xylostella* by Douris et al., in 2016 elaborated that the CRISPR/Cas9 system can convincingly elucidate the molecular mode of action of the *CHS1*-inhibiting bioactive molecules. The same research group also demonstrated a mutation in the *PxCHS1* gene which was responsible for the resistance against benzoylureas (BPUs) in *P. xylostella* [79].

The homeotic gene abdominal-A of the Diamondback Moth, *Plutella xylostella* (L.) [80], and the Fall Armyworm (FAW), *Spodoptera frugiperda* [81], was targeted using the CRISPR/Cas9 system. The double sex (*PxDSX*) gene of the *P. xylostella* responsible for sex determination in the insect was targeted using the CRISPR/Cas9 system which led to the altered expression of the sex-biased genes [82]. A disruption in the normal functioning of the vital genes by an induced series of deletions and insertions in the genomic locus using the CRISPR/Cas9 system provided the groundwork for the development of new tools for pest management. In a report on Lepidopteran pests, *Spodoptera litura* CRISPR/Cas9-mediated mutagenesis targeted the *Slabd-A* gene (*S. litura* abdominal-A). The *Slabd-A* gene has a central role in abdominal segmentation and segment identity determination in insects. Thus, *Slabd-A* is involved in embryonic development, and therefore, targeting this gene for genome manipulation resulted in ectopic pigmentation and anomalous segmentation [83]. A study on *S. litura* demonstrated that *SlitBLOS2* gene knockout led to a complete disappearance of the white spots and yellow strips on the larval integument, and thus, this provided a marker gene for functional studies and pest control strategies [84].

The most essential odor-binding proteins are PBPs (Pheromone Binding Proteins) [85]. To demonstrate the function of the PBP gene in *S. litura*, the PBP3 gene (SlitPBP3) was

mutated using the CRISPR/Cas9 system. The response of the obtained *SlitPBP3* mutants was compared with that of the wild-type individuals, and the mutants exhibited a reduced response to sex pheromone components [86]. The knockouts of the olfactory receptor co-receptor (*Orco*) gene of *Spodoptera littoralis* were acquired using the CRISPR/Cas9 system. These knockouts were not able to respond to sex pheromones and plant odors [87]. Recently, a study was conducted which targeted three genes in *S. frugiperda*. The three target genes included two marker genes tryptophan 2, 3-dioxygenase (*TO*), the biogenesis of lysosome-related organelles complex 1 subunit 2 (*BLOS2*), and one developmental gene, *E93* (a key ecdysone-induced TF promoting development in adult insects). This study envisioned the method using multiple sgRNA injections, and the results have a possible use in functional gene characterization and high-throughput functional genomics screening for an in-depth understanding of the complex mechanisms regulating the crucial pathways of FAW and other invasive pests. In *S. exigua*, the functional validation of Seα6-KO (Homozygous strain) was performed using the CRISPR/Cas9 system [88].

### 1.3.3. *Dendrolimus punctatus*

*Dendrolimus punctatus*, which is commonly called the Pine Caterpillar Moth, has a devastating effect on the resin production of Southeast Asia and China. The embryo development process is well known to have an association with the *Dpwnt-1* gene for early body planning. A CRISPR/Cas9-mediated mutation in the *DpWnt-1* gene in *D. punctatus* greatly altered the gene expression, which led to abnormal growth and deformed organ development. In this study, a mixture of EGFP expressing gene cassette and sgRNA/Cas9 mRNA was used for the embryonic injection [89].

### 1.3.4. *Cydia pomonella*

The Codling Moth, *Cydia pomonella*, is a major pome fruit pest. The odorant receptor of *C. pomonella* (*CpomOR1*) was targeted using the CRISPR/Cas9 system. The *CpomOR1* gene expresses at a high quantity in the antennae which encodes the codlemone receptor (an odorant receptor). In this study, early-stage eggs were injected with sgRNA and Cas9 mRNA. The emerging female with the knockout *CpomOR1* gene was allowed to mate with a normal male, and the researchers observed that their fecundity and fertility were affected as they produced non-viable eggs [90].

### 1.3.5. *Ostrinia furnacalis* (*Lepidoptera*: *Pyralidae*)

It is a major agricultural pest in Asia that primarily feeds on corn crops. The CRISPR/Cas9 system was used to understand the function of a developmental gene. The knockout study of the *Ago1* gene was performed in *O. furnacalis* (*OfAgo1*). In this study, an sgRNA/Cas9 mRNA mix was injected in newly laid eggs. In the hatched larvae, the loss of function of the *OfAgo1* gene disrupted the cuticle pigment of the seventh abdominal and third thoracic segments. This finding reveals the role of this gene for cuticle pigmentation [91]. Future studies in respect of pest control can be performed by following the insect rearing and transfection methods for this insect.

### 1.3.6. *Agrotis ipsilon*

It is also known that the Black Cutworm is a harmful pest that feeds on almost all of the important vegetables and grains. The CRISPR/Cas9 system was used for developing the mutants for the *tyrosine hydroxylase* (*AiTH*) gene. A mixture of sgRNA and Ca9 mRNA was microinjected in fresh eggs, and they were incubated until they were ready for hatching. Furthermore, the sgRNA and Ca9 mRNA-injected egg groups were observed by the narrowing of the eggshell that led to a failure in the hatching in most of the eggs. Additionally, if they did hatch, the larva was severely dehydrated and died after one day [92].

### 1.3.7. *Hyphantria cunea*

*Hyphantria cunea* (Fall Webworm), Drury, is an invasive pest that feeds on important fruits and crop fields worldwide. Li et al. targeted double sex (*Hcdsx*) gene using the CRISPR/Cas9 technology, and the mutants were observed to have abnormal external genitalia and incomplete sex reversal phenotypes. This led to a reduced sex-specific fecundity. In the same report, the alternative splicing pattern of the *Hcdsx* gene was also altered using the CRISPR/Cas9 system, and the alterations in the splicing pattern changed the downstream gene expression of *vg1*, *vg2* (encoding vitellogenin) and pheromone binding protein 1, which in turn led to the development of sex-specific sterile phenotypes in the *Hcdsx* mutants [93].

### 1.3.8. *Mythimna separata*

*Mythimna separata* (Walker) is a phytophagous pest that has been recently targeted for GE for the first time. Tang et al. in 2022 optimized the application of the CRISPR/Cas9 system by targeting the NPC1b gene. The NPC1 family proteins are involved in intestinal absorption and sterol trafficking. The eggs were microinjected with Cas9 protein and sgRNA (RNP complex). The optimized methods from this study can be exploited for the designing of novel pest control strategies [94].

### *1.4. Hemiptera*

The order Hemiptera includes insects of the three major suborders Auchenorrhyncha (Spittlebugs, Cicadas, Planthoppers and Leafhoppers), Heteroptera and Sternorrhyncha (Whiteflies and Aphids) which feed almost entirely on plant sap. These insects have adapted to a wide range of diets. Hemiptera is the insect order with the common name of bugs (as flies represent the Diptera). The main features of Hemipteran insects are the sucking and piercing mouthparts. Various control strategies have been applied to control the hemipteran insects. CRISPR/Cas9-mediated genetic alterations have shown remarkable potential in providing a simple and heritable approach in the field of pest management and functional studies of the Hemipteran pests.

### 1.4.1. *Nilaparvata lugen*

*Nilaparvata lugens* which are also known as the Brown Planthopper is a highly destructive insect pest in Asia, and given that it is highly fertile, it causes great agricultural losses by sap sucking rice plants and also by transmitting various viruses. For removing the constraints in the field of functional genomic studies, the CRISPR/Cas9 system was used to target two eye pigmentation genes, namely, the *cinnabar* gene (*Nl-cn*) and the *white* gene (*Nl-w*) of *N. lugens*. These genes were knocked out and further validated by an RNAi-based knockdown analysis. This capability of introducing precise genetic alterations provides alternative means for understanding the gene function and establishing new approaches for pest control in this non-model pest [95].

### 1.4.2. *Diaphorina citri*, *Homalodisca vitripennis*, *Bemisia argentifolii* and *Bemisia tabaci*

*Diaphorina citri* (Asian citrus psyllid) acts as a vector for *Candidatus Liberibacter asiaticus* (CLas), a pathogenic bacterium. When the psyllid feeds on the citrus plants, it causes the transmission of a disease that is known as Huanglongbing (HLB). Hunter et al. in 2018 made an effort to improve the delivery of the CRISPR components by the addition of Branched Amphiphilic Peptide Capsules, BAPC. Injecting the gene-editing components near the insect ovaries led to the production of a heritable germline with the edited genome in the subsequent generations. It has bypassed the requirement of microinjecting the eggs [96]. Later, in 2019, Hunter et al. utilized this BAPC-assisted-CRISPR–Cas9 method for gene targeting and gene editing in insect nymphs and adult insects (Psyllids—*Diaphorina citri*; Leafhoppers—*Homalodisca vitripennis*; Whitefly—*Bemisia argentifolii*). The two genes that were targeted for the knockouts were the *Vermillion*, *Vm*, and the *thioredoxin* gene, *TXT*. The knockouts changed their eye color and physiology, respectively. The insects that were

selected for this study are a threat to food security as they transmit pathogenic bacteria and viruses to plants worldwide. The idea behind the development of this strategy was to change the vectors into non-vectors. The results were them having a reduced lifespan, slower development, reduced fecundity and it changed their eye phenotype. BAPC-assisted CRISPR delivery reformed the approaches to protect food crops from different pathogens and insect vectors [97]. Recently, in 2020, a study was performed on *B. tabaci,* demonstrating the development of a CRISPR–Cas9 based gene editing technique in which the vitellogenic adult females were injected with the CRISPR/Cas9 components rather than the embryos. Here, the Cas9 protein was fused with an ovary-targeting peptide ligand ("BtKV"), leading to efficient and heritable gene editing in the genome of the offspring. These adult injections were easy to administer and there was no need for specialized tools [98] (Figure 2C).

### 1.4.3. *Euschistus heros*

In *Euschistus heros,* the Neotropical brown stink bug, three genes *yellow (yel)*, *tyrosine hydroxylase (th)* and *abnormal wing disc (awd)* were targeted through RNAi to study the gene function. The *awd* gene knockdown insects had deformed wings and the *th* gene targeted insects had a lighter cuticle pigmentation. Thus, the RNAi-mediated targeting revealed that distinct malformed phenotypes are linked to both of the genes, but no distinct phenotype was observed for the *yel* gene. Further, to understand the function of this gene in this insect, the CRISPR/Cas9-mediated knockout technique was developed. However, still there was no distinct phenotype differences in *yel* gene mutant insect when they were compared to normal insects. For this study, the *yel* gene-targeting RNP complex was microinjected in the eggs, and the hatched larvae were observed for the altered phenotype studies [99].

### *1.5. Coleoptera*

Coleopteran comprise of the largest order of insecta. In this order, the most common pests of the stored crop products are found. The adult insect of coleoptra have forewings that are modified as hard elytra. They include beetles which inhabit an ample variety of habitats.

### 1.5.1. *Tribolium castaneum*

*Tribolium castaneum*, which is also called the Red Flour Beetle, is a pest of stored grains and other agricultural products. They lower the nutritional value of the crop. Their secretions possess carcinogens like benzoquinone. The availability of the genomic database of *T. castaneum* [100,101] has upgraded the *T. castaneum* eradication methods from traditional fumigation to the genome editing-based methods. These new methodologies can expand pest management and diminish the damage that is caused to the environment. The CRISPR/Cas9 system was applied to *T. castaneum* for the first time [102]. In this report, Gilles et al. established that a mutated *E-cadherin* gene led to dorsal closure defects. Gilles' group also attained a homology-directed knockin using the CRISPR system, which turned out to be very efficient [103]. Subsequently, this GE system can be used for various species, providing basic information for launching CRISPR-centered transgenic techniques. This could minimize the development-associated voids between the non-model organisms and the model organisms [104].

### 1.5.2. *Leptinotarsa decemlineata*

In January 2020, the CRISPR/Cas9 system was used for the first time in the Colorado potato beetle (CPB), *Leptinotarsa decemlineata.* Initially, the *vestigial* gene (*vest*) was functionally characterized in CPB through RNAi, which was followed by the establishment of the CRISPR/Cas9 protocol in CPB for the mutagenesis study. The RNAi-induced phenotype of deformed wings reappeared in the *vest* gene mutants, and this was also developed using the CRISPR/Cas9 system. In this mutagenesis study, the RNP complex was microinjected in <1 h old eggs, and they were further incubated until they were ready for hatching.

Altogether, this report provided an improved environmentally friendly pest management methodology [105].

### 1.6. Orthoptera

Orthoptera possess chewing and biting mouthparts, and they damage the crop by biting off different plant parts. The type of damage that is caused by Orthoptera is the direct damage to the plant, and it is not indirect damage. The damage is visible, and it is similar to the damage that is caused by grubs and caterpillars.

### Locusta migratoria

*Locusta migratoria* is also called the locust, and its breakouts pose an unprecedented threat to agriculture, and thus, to the livelihood and food supply of millions of people. The classical eradication policies to deal with locust breakouts depend on chemical insecticides, which take a toll on the farmers' pockets as well as on the environment. Locusts are considered to be important by scientists for molecular studies also. The developmental synchronization among the migration, social behavior and copulation patterns of the insect act as the foundation in determining the density of the locusts [106]. The insects' behavioral responses such as mating, feeding, foraging and spawning are directed by the brain. The pheromones are perceived by the peripheral tissues, processed in the nerve tissue and they are passed on to the sensory organs like the olfactory and others. Finally, the brain receives and integrates these signals, thus, the olfactory plays a vital role in insects. To obtain a better insight on the functional genes of *Locusta migratoria,* the odorant receptor co-receptor (*Orco*) gene was targeted by the CRISPR/Cas9 system, and consequently, high mutation rates were achieved. The results suggested novel strategies for locust control, and also, the frame of this study may be used in controlling other insects of the same clade such as crickets [107].

### 1.7. Acarina

The family Acaridae is a pest of cereal products and stored grains, having blunt, toothed chelicerae, enabling them to gouge and scrape the plant material.

### Tetranychus urticae

A successful genetic transformation or genome editing is yet to be established for chelicerates. Some tick and mite species within this group are of great economic and agricultural importance. Accessibility to a well-developed genome-editing tool would be a significant improvement in this area. The spider mite *Tetranychus urticae* is a polyphagous pest. They transmit viruses and travel by water and wind. The insecticides such as pyrimidinamines, quinazolines, pyrazoles and pyridazinones block the quinone binding receptor of mitochondrial respiratory complex I in the *T. urticae*. These chemicals are categorized as mitochondrial electron transport inhibitors (METIs). In the *T. urticae* strain which are resistant to these, the METIs were identified with a mutation in the *PSST* homolog of the complex I [108]. It was demonstrated that substitution in the H92R amino acid of the *PSST* homolog is responsible for their resistance against pyridaben. In 2020, *Phytoene desaturase* gene mutants of *T. urticae* were developed through an injection of RNP complex into the ovary of virgin females. The mutation caused albino phenotypes in subsequent progenies. This research provided an impetus for the genetic transformation of chelicerates and paved the way for functional studies using the CRISPR/Cas9 system in *T. urticae* [109].

## 2. The CRISPR/Cas9 System in Pest Management: Challenges and Future Prospects

In the past decade, the CRISPR/Cas9 system has emerged as an elegant and affordable genetic technique, and it is expected to be extensively applied in pest control applications for crop improvement in the near future [88]. Even after the rapid progression in the field of the CRISPR/Cas9-based genome editing methods, researchers are facing challenges such as off-target effects, efficient delivery methods and genome editing efficiency. Despite

these challenges, the CRISPR/Cas9 system has the potential for revolutionizing the field of agriculture. Now, there is a need to focus on the improvement of the CRISPR/Cas9-based pest management technologies such as combining the SIT with the CRISPR/Cas9 system, which has great pest control capabilities as it targets the reproductive fitness of them (Figure 2B). Similarly, the utilization of the CRISPR/Cas9 system for transcription regulation could be a promising technique, but the genetically modified organism in such cases will not be "transgene-free" as it will carry a transgene for dead Cas9 (dCas9) and gRNA expression cassette. Recently, one more novel approach has been explored which involves the transportation of the injected RNP complex which is assisted by the ovary-targeting peptide ligand or BAPC in nymphs and adult flies. Using this method, the CRISPR components can be directly injected near the ovary of an adult female fly, thus making it more convenient when it is compared to the microinjections in the eggs or embryos (Figure 2C). This approach has a lot of scope, and this similar concept can be applied in other insect orders for pest management.

## 3. Discussion

Plant pests are likely to have considerable effects on the crop distribution, density, yield and crop spread as a result of climate change. Pesticides act as a vital tool for pest management and their consumption across the world is reported to be over four million tons per year [108]. As per the FAO, crop destruction by insect pests leads to the starvation of millions of people. Over and above one thousand pesticides are existing presently in the market including microbial formulations, chemical ones, semi-chemical ones, herbal ones and others. The persistence of pesticides in the environment poses a worldwide threat to human well-being and ecological systems. Thus, the development of alternative pest management strategies is the need of today. The development of new integrated pest management strategies is required, which have zero side effects on the non-target insect population and lead to effective pest control which further can ensure crop improvement and food safety.

**Table 1.** CRISPR/Cas9-based genome editing in various insects.

| Insect Species | Target Gene | Accession Number | Genetic Trait | Mutation Type | Delivery of CRISPR Components | Findings | References |
|---|---|---|---|---|---|---|---|
| *Drosophila melanogaster* | *yellow, rosy* | NM_057444.3, NM_079613.3 | Pigmentation and Mating | Knockout, Knockin | Plasmid | This was the first report using the CRISPR/Cas9 system to mediate efficient genome engineering in Drosophila. | Gratz et al., 2013 |
| | *yellow, white* | NM_057444.3, NM_079613.3 | Pigmentation and Mating | Knockout | Cas9 mRNA and sgRNA | sgRNA concentration-dependant knockout was shown for *yellow* gene, and highly efficient and varied genome editing efficiencies were shown by different sgRNAs. | Bassett et al., 2013 |
| | *yellow* | NM_057444.3 | Pigmentation and Mating | Knockout | Cas9 mRNA and gRNA | This report used the approach of targeting multiple genes with different sgRNAs, and it attained a remarkably effective targeted mutagenesis. | Yu et al., 2013 |
| | *Ast, Eh, capa, Ccap, Crz, npf, Mip, mir-219, mir-315, white* | NM_001300582.1, NM_079662.3, NM_079828.3, NM_001275917.2, NM_079626.3, NM_080493.3, NM_140714.4, NR_048289.1, NR_048297.1, X76202.1 | | Knockout | Plasmid | To obtain a Cas9–sgRNA complex for achieving targeted mutagenesis, two transgene vectors harboring expression cassettes for Cas9 and sgRNA were delivered. | Kondo and Ueda, 2013 |
| | *rosy, DSH3PX1* | NM_079613.3, NM_140091.4 | Pigmentation and Mating | Knockout, Knockin | Plasmid | Executed efficient and complex genomic manipulations using CRISPR/Cas9-mediated HDR. | Gratz et al., 2014 |
| | *ebony, yellow, wingless, wnt* | NM_079707.4, NM_057444.3, NM_078778.5 | Segmentation | Knockout, Knockin | Plasmid | Different promoters were used to drive sgRNA expression, and based on promoter properties, different patterns of expression were observed. | Port et al., 2014 |
| | *EGFP, mRFP* | | Chromogenic fluorophores | Knockout | Plasmid | Induction of mutations by injection of an sgRNA into Vasa-Cas9 transgenic fly embryos. | Sebo et al., 2014 |
| | *white, piwi* | NM_057439.2, NM_001298896.1 | Pigmentation and Expression of group of small RNA | Knockout, Knockin | Plasmid | Used Cas9 nickase and sgRNA pairs to prevent off-target effects during the generation of indel mutants. | Ren et al., 2014 [110] |
| | *ms(3)k81, white, yellow* | NM_143253.2, NM_057439.2, NM_057444.3 | Pigmentation | Knockout, Knockin | Plasmid | CRISPR mediated genome editing was shown in Drosophila. | Xue et al., 2014a |

**Table 1.** *Cont.*

| Insect Species | Target Gene | Accession Number | Genetic Trait | Mutation Type | Delivery of CRISPR Components | Findings | References |
|---|---|---|---|---|---|---|---|
| *Drosophila melanogaster* | *yellow, notch, bam, nos, ms(3)k81, cid* | NM_057444.3, NM_001258581.2, NM_057452.4, NM_057310.4, NM_143253.2, NM_079006.4 | Physiology | Knockout | Plasmid | A CRISPR/Cas9-mediated conditional mutagenesis system combined with tissue-specific expression of Cas9 was used to temporally and spatially inhibit gene expression. | Xue et al., 2014b |
| | *salm* | NM_164966.3 | Zinc Finger Transcriptional Repressor | Knockin | mRNA, transgene | For flexible modification of fly genome, a two-step method was proposed. | Zhang X. et al., 2014 |
| | *ebony, yellow, vermilion* | NM_079707.4, NM_057444.3, NM_078558.3 | Pigmentation | Knockout, Knockin | Plasmid, transgene | Donor template and sgRNA plasmids were injected into Cas9 transgenic embryos in Drosophila. | Ren et al., 2014b |
| | *ebony, yellow, white* | NM_079707.4, NM_057444.3, NM_057439.2 | Pigmentation | Knockout, Knockin | Plasmid, transgene | A bicistronic Cas9/sgRNA vector was constructed which enhanced the efficiency of gene targeting. | Gokcezade et al., 2014 |
| | *ebony, yellow, wg, wls, Lis1, Se* | NM_079707.4, NM_057444.3, NM_078778.5, NM_140188.4, NM_057812.5, NM_139978.4, NM_057812.5 | Pigmentation and Physiology | Knockout, Knockin | Plasmid, transgene | Non-transgenic individuals exhibited lesseficient knockin than transgenic individuals did. | Port et al., 2015 [111] |
| | *yellow* | NM_057444.3 | Pigmentation | Knockin | Transgene | Heterozygous recessive mutation was converted to homozygous loss of function mutations utilizing mutagenic chain reaction (MCR) technology in *Drosophila.* | Gantz and Bier, 2015 |
| | *Dα6* | NM_164874.3 | Insecticide resistance | Knockin | Plasmid | The G275E mutation of the *nAChR Dα6* subunit is directly related to Spinosad resistance. | Zimmer et al., 2016 |
| | *LUBEL* | NM_001273232.2 | Growth and Development | Knockout | Plasmid | Flies with *LUBEL* mutations exhibited reduced survival and defective climbing in response to heat. | Asaoka et al., 2016 |
| | *Scsα* | NM_079181.4 | Growth and Development | Knockout | Plasmid | Mutant flies could not produce sufficient energy to promote normal growth. | Quan et al., 2017 |
| | *clamp* | NM_136293.4 | Sex Specific | Knockout | Plasmid | The expression of a sex-specific gene was regulated by an essential transcription factor. | Urban et al., 2016 |
| | *chameau, CG4221, CG5961* | NM_135273.5, NM_141949.4 | | Knockin | mRNA | HDR-mediated genome modifications efficiency was tested, and a problem associated with "ends-in" recombination was resolved. | Yu et al., 2014 |
| | *fdl* | NM_165908.2 | | Knockout | Plasmid | Capability of CRISPR/Cas9 system for analysing or manipulating protein glycosylation pathways. | Mabashi-Asazuma et al., 2015 [112] |
| | *mod(mdg4)* | NM_163878.2 | | Knockout | Plasmid | Validation of a functional gene involved in trans-splicing that influenced the development in flies. | Gao et al., 2015 [113] |
| | *act5C, lig4, mus308* | NM_167053.2, NM_132679.3, L76559.1 | | Knockout, Knockin | Plasmid, transgene | Offered a comprehensive technique for genome editing in Drosophila S2 cells. | Kunzelmann et al., 2016 [114] |
| | *yellow, white, tan* | NM_057444.3, NM_057439.2, NM_132315.1 | Pigmentation | Knockin | Plasmid, transgene | Proposed a new process of attaining single or multiple allelic substitutions. | Lamb et al., 2016 |
| | *wntless* | NM_140188.4 | Growth and development | Knockout | Plasmid | A complex of tRNA–sgRNA was proposed to amplify the cleavage efficiency of the Cpf1 and Cas9 nucleases. | Port and Bullock, 2016 |
| | *TpnC* | NM_078895.4 | Growth and development | Knockout | Plasmid | Confirmed that the myofibril assembly is related to *TpnC* gene. | Chechenova et al., 2017 |
| | *Alk* | NM_144343.3 | Growth and development | Knockout | Plasmid | Revealed that transcription factors can affect *Alk* gene expression by establishing mutations in Alk enhancer regions. | Mendoza-Garcia et al., 2017 |
| *Drosophila suzukii* | *white (w-)* | NM_057439.2 | Pigmentation | Knockout | Plasmid | Absence of mating and copulation failure was reported. The mutation also caused pigmentation deficiency in testis sheath, which could be a probable reason for copulation failure. | Yan et al., 2020 |
| | *white, Sxl* | NM_057439.2, XM_017083263.2 | Sex determination | Knockout | Plasmid | *Sxl* gene was proved as excellent gene to suppress the population growth of this destructive pest. | Li and Scott, 2016 |
| | *DsRed* (red fluorescence protein) | | | knockin | Plasmid, transgene | The enhancer/promoter of the spermatogenesis-*specific beta-2-tubulin (β2t)* gene was used for expression of fluorescent proteins or effector molecules in testes of pests, and this providing basis for reproductive biology studies sexing and monitoring. | Ahmed, H. M. et al., 2019 |
| *Drosophila subobscura* | *yellow, white* | XM_034814491.1, XM_034808177.1 | Pigmentation | Knockout | mRNA | Gene functions were analyzed in a non-model Drosophila species. | Tanaka et al., 2016 [115] |
| *Anastrepha ludens* | *Astra-2* | EU024509.1 | Sex determination | Knockout | RNP complex | The mutation caused sterility, thus, the target gene was proposed for helping in pest control. | Li et al., 2019 |
| *Bactrocera dorsalis* | *White* and *transformer* | AY055817.1, KP342062.1 | Sex determination and reproduction | Knockout | RNP complex | CRISPR/Cas9 mediated mutation of *white* and *transformer* genes caused various phenotypic effects. | Zhao et al., 2018 |

**Table 1.** *Cont.*

| Insect Species | Target Gene | Accession Number | Genetic Trait | Mutation Type | Delivery of CRISPR Components | Findings | References |
|---|---|---|---|---|---|---|---|
| *Ceratitis capitata* | • eGFP_gRNA2<br>• eGFP_gRNA2 and 1 mM Scr7<br>• eGFP_gRNA2b–Cas9 complexes with ssODN_BFP donor template | | Homology directed repair | knockin | RNP complex and a single-stranded oligo donor | Conversion of eGFP-to-BFP was demonstrated for establishing an efficient HDR through CRISPR-based genome editing. | Aumann, R. A. et al., 2018 |
| | • *white* eye (we)<br>• *paired* gene (Ccprd) | X89933.1, XM_020858622.2 | Segmentation | Knockout | RNP complex | A simple and highly efficient RNP complex-based genome editing approach was reported with the details of designing and preparation. | Meccariello, A. et al., 2017 |
| *Helicoverpa armigera* | *NPC1b* | MK555324.1 | Growth and dietary uptake of Cholesterol | Knockout | RNP complex | NPC1b is vital for the growth and dietary cholesterol uptake. Thus, a novel pest-management technique can be developed using NPC1b as an insecticidal target. | Zheng, J. C. et al., 2020 |
| | *HaCad* | JX23382.1 | cell–cell adhesion | Knockout | sgRNAs and Cas9 mRNA | sgRNAs and Cas9 mRNA were injected into the fresh eggs, and a high editing efficiency of the HaCad locus was achieved. | Wang, J. et al., 2016 |
| | *HaABCA2* | KP259911.1 | Regulation of enzymes | Knockout | Cas9 mRNA and sgRNA | The knockout of *HaABCA2* confirmed the role of *HaABCA2* in mediating toxicity of both Cry2Aa and Cry2Ab against *H. armigera*. | Wang, J. et al., 2017 |
| | *odorant receptor 16 (OR16)* | KF768670.1 | Olfaction | Knockout | Cas9 mRNA + sgRNA and RNP complex | The results represent the basis for novel olfactory-based strategies of pest population control. | Chang, H. et al., 2017 |
| | *white, ok, brown,* and *scarlet* | XM_021344759.2, KU754490.1, KU754480.1, KU754478.1 | Pigmentation | Knockout | Cas9 mRNA | The report represented differential distribution of eye pigments in the mutants; this finding may be helpful in elucidation of biosynthetic pathway. | Khan, S. A.,et al., 2017 |
| | cluster of nine P450 genes | KM016735.1, R095600.1, KM016739.1, KM016740.1, DQ256407.1, KM016743.1, KM016741 | Regulation of detoxifying enzymes | Knockout | Cas9 protein and multiple sgRNAs | The report identified the key players in the insecticide metabolism. | Wang, H. et al., 2018 |
| *Plutella xylostella* | *Pxabd-A* | XM_011570968.3 | Body segmentation | Knockout | Cas9 mRNA | CRISPR/Cas9 was used to target genes in *P. xylostella* for the first time which provided new ideas for pest control. | Huang et al., 2016 |
| | *Pxdsx* | XM_048630440.1 | Sex determination | Knockout | Microinjection of RNP complex | The results showed CRISPR/Cas9 system led to altered expression of sex biased genes. | Wang et al., 2019 |
| | *PxCHS1* | AB271784.1 | Development | Knockout | Plasmid | Description of the resistance management strategies for insect pests, it and explained the MoA behind the resistance using CRISPR/Cas9 system. | Douris et al., 2016 |
| | *LW* | | Locomotion | Knockout | RNP complex | The results showed weaker phototaxis and reduced locomotion, thus making it a helpful method for pest control | Chen et al., 2021 [116] |
| *Spodoptera frugiperda* | *Sfabd-A* | MH541836.1 | Body segmentation | Knockout | RNP complex | The results showed that gene function validation and the understanding of resistance mechanism can be performed using CRISPR/Cas9 system which can lead to the development of novel pest management approaches. | Wu et al., 2018 |
| | • BLOS2<br>• E93<br>• TO | XM_035582273.2 XM_050696092.1 XM_050696079.1 | Growth and development | Knockout | Cas9 protein and multiple sgRNAs | The developed mutants were helpful to understand the crucial pathways of *S. frugiperda* and the strategy can also applied for other invasive pests. | Zhu, G. H. et al., 2020 [117] |
| *Spodoptera litura* | *Slabd-A* | GCA_002706865.1 | Body segmentation | Knockout | Cas9 mRNA and sgRNA | The direct injection of Cas9-coding mRNA and *Slabd-A*-specific sgRNA into the embryos of the *S. litura* led to the induction of the typical *abd-A* deficient phenotypes showing irregular segmentation and unusual pigmentation at the larval stage. | Bi, H. L. et al., 2016 |
| | *SlitBLOS2* | XM_022977403.1 | Molecular marker | Knockout | Cas9 mRNA and sgRNA | The study demonstrated that SlitBLOS2 has a role in the coloration of the integuments, and thus, it provided a marker gene for functional studies and pest control strategies. | Zhu, G. H. et al., 2017 |
| *Spodoptera littoralis* | *SlitOrco* | | Olfaction | Knockout | mRNA | The *Orco* gene was investigated in the insect *Spodoptera littoralis*. The results were helpful in making a pest control strategy and in gene function analysis. | Koutroumpa et al., 2016 |
| *Spodoptera exigua* | *Seα6* | MN714701.1 | | Knockout | RNP complex | The study demonstrated that knocked-out Seα6 was highly resistant to insecticides. | Zuo et al., 2020 |
| *Dendrolimus punctatus* | *DpWnt-1* | KU640201.1 | Development and segmentation | Knockout | mRNA | Proved the necessity of DpWnt-1 signaling in appendage development and anterior segmentation. | Liu H. et al., 2017 |
| *Cydia pomonella* | *CpomOR1* | FJ385021.1 | Olfaction | Knockout | Cas9 mRNA and sgRNA | The report demonstrated mutation in the *CpomOR1* gene via CRISPR/Cas9 affected the egg production and viability in the insect. | Garczynski, S. F. et al., 2017 |

**Table 1.** *Cont.*

| Insect Species | Target Gene | Accession Number | Genetic Trait | Mutation Type | Delivery of CRISPR Components | Findings | References |
|---|---|---|---|---|---|---|---|
| *Ostrinia furnacalis* | *OfAgo1* | | Growth and development | knockout | sgRNA and Cas9 mRNA | Mutation in *OfAgo1* gene through CRISPR/Cas9 technology caused cuticle disruption. | You et al., 2019 |
| *Agrotis ipsilon* | *AiTH* | | Growth and development | Knockout | sgRNA and Cas9 mRNA | The *AiTH* gene knockout by CRISPR/Cas9 caused narrowing in the egg shell. | Yang et al., 2018 |
| *Hyphantria cunea* | Hcdsx | | Reproduction | Knockout | sgRNA and Cas9 mRNA | Knocked-out *Hcdsx* gene by CRISPR/Cas9 caused sex-specific sterility, thus making it a pest control method. | Li et al., 2020 |
| *Mythimna separata* | *NPC1b* | MZ209049.1 | Intestinal absorption and sterol trafficking | Knockout | RNP complex | Knockout of *NPC1b* can hamper nutrient absorption. | Tang et al. 2022 |
| *Nilaparvata Lugens* | *Nl-cn* and *Nl-w* | MH105806.1 | Pigmentation | Knockout | Cas9 mRNA and sgRNA | Two genes for eye pigmentation were targeted using CRISPR/Cas9, and the results paved path for gene-function interrogation. | Xue. et al., 2018 |
| *Diaphorina citri* | *ACP-TRX-2* | XM_026831570.1 | Physiology | Knockout | BAPC-assisted delivery of CRISPR components | The method incorporated BAPC-assisted delivery of CRISPR/Cas9 into nymphs and adults, thus resulting an innovative breakthrough in gene editing, it and has shown a significant improvement over efforts using injection of eggs. | Hunter et al., 2018 |
| *Diaphorina citriHomalodisca vitripennis, Bemisia argentifolii* | *Thioredoxin* and *Vermillion* | XM_046819472.1 | Physiology andEye color | Knockout | BAPC, plasmid, dsRNA | The BAPC-assisted delivery system developed gene editing methods across the all hemipteran pests by permitting the use of nymphs and adults. BAPC-assisted CRISPR delivery transformed the approaches to protect food crops from different pathogens and insect vectors. | Hunteret al., 2019 |
| *Bemisia tabaci* | *white* | XM_019053144.1 | Pigmentation | Knockout | SgRNA + Cas9 protein fused with overy targeting peptide ligand (BtKV) | The method has significantly expanded the capability of CRISPR techniques for whitefly research. | Heu et al., 2020 |
| *Euschistus heros* | *abnormal wing disc (awd), tyrosine hydroxylase (th) and yellow (yel)* | NP_001119625.1, XP_008182999.1, XP_001948479.1 | Body segmentation and pattern | Knockdown and knockout | dsRNA, RNP complex | Use of RNAi and CRISPR/Cas9 techniques for managing insect pests. | Cagliari et al., 2020 |
| *Tribolium castaneum* | *Tribolium E-cadherin* | XM_961215.3 | Dorsal closure defect | Knockout | Plasmid | *Tribolium E-cadherin* gene was targeted for knockout study. | Gilles et al., 2015 |
| *Leptinotarsa decemlineata* | *vestigial gene (vest)* | XM_023168389.1 | Growth and development | Knockout | RNP complex | Functionally characterized *vest* gene and CRISPR/Cas9 protocol was established for mutagenesis. | Gui, S. et al., 2020 |
| *Locusta migratoria* | *Orco* | JN989549.1 | Olfaction | Knockout | mRNA | Functional genetic studies of locusts by generation of loss-of-function mutation for managing insect pests. | Li Y. et al., 2016 |
| *Tetranychus urticae* | *PSST* | KX806605.1 | | Knockout | Plasmid | Substitution in the H92R amino acid of the PSST homolog was related to pyridaben resistance and the mutation into the *Drosophila* PSST homolog using CRISPR/Cas9 genome-editing tools. | Bajda et al., 2017 |
| | *phytoene desaturase* | MF167355.1 | | Knockout | RNP complex | Induction of two mutagenetic events using CRISPR/Cas9 providing basis for functional studies. | Dermauw, W. et al., 2020 |

**Author Contributions:** S.S., S.R., S.P., G.S., S.K.U., G.M. and P.C.V. have conceived and planned this article and all the authors have reviewed it. S.S. and S.R. wrote the first draft; all authors contributed to and approved the final manuscript. All authors have read and agreed to the published version of the manuscript.

**Funding:** This work was supported by SERB Department of Science and Technology (Govt. of India) funded project under the grand agreement CRG2020001095.

**Institutional Review Board Statement:** Approved for publication. Institutional Manuscript ID no. CSIR-NBRI_MS/2022/07/14.

**Informed Consent Statement:** Not applicable.

**Data Availability Statement:** Not applicable.

**Acknowledgments:** P.C.V. and G.M. are thankful to the SERB, Department of Science & Technology (DST), Govt. of India for the financial support (CRG/2020/001095). The authors would like to thank CSIR, National Botanical Research Institute, Lucknow (U.P.), India for necessary support and facilities.

**Conflicts of Interest:** Authors have no conflict of interest to declare.

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
