# Peer review of "CRISPR/Cas9 for Insect Pests Management: A Comprehensive Review of Advances and Applications"

_agriculture, doi:10.3390/agriculture12111896_

Round 1
Reviewer 1 Report
Singh et al. provided a review on the utilization of CRISPR/Cas9 for insect pest management. Though the topic is of current interest, the manuscript needs a substantial revision to be published. I have mentioned a few points of the manuscript that can be considered by the authors.
-I suggest authors add a few lines about the benefit of the manuscript in the abstract. They can add how the presented data will be beneficial for further research work in this direction. The novelty of this manuscript should be highlighted in both the abstract and the introduction.
-Line 26: Remove ‘2. Materials and Methods’.
-The English language needs to be corrected by a native English speaker (Line 42, 45, 47, and so on).
-Please add the aim of the manuscript in the introduction before starting 1.1.
-In 1.2.2, either please add a few more studies or elaborate on the study of Li and Handler 2019 in detail. Same comment is applicable for 1.2.3; 1.3.3; 1.3.4; 1.3.5; 1.3.6; 1.5.2. Please describe the results of the study in 1.4.3.
-Please describe some information on 1.6. and 1.7 as it was provided for 1.2, 1.3, 1.4 etc. to maintain the symmetry.
-Please modify Line 422.
-Before conclusions, it will be better to add a separate heading describing the challenges in the application of the CRISPR technique for insect pest management.
I do believe that the manuscript can be accepted once the authors address the mentioned points and enrich the manuscript with crucial information.

Author Response
Replies To Comments By Reviewer 1
We would like to thank the reviewer for their kind criticism and time. We believe the incorporated changes will answer the queries, bring clarity and will satisfied the reviewer. We are keen on improving our article kindly, feel free to share better ideas.
Comment 1 -I suggest authors to add few lines about the benefit of the manuscript in the abstract. They can add that how the presented data will be beneficial for the further research work in this direction. The novelty of this manuscript should be highlighted in both the abstract and introduction. –
Response- Suggested change has been incorporated (line 27 to 30)
Comment 2 -Line 26: Remove ‘2. Materials and Methods’.
Response- Suggested change has been incorporated
Comment 3 -The English language needs to be corrected by a native English speaker (Line 42, 45, 47 and so on).
Response- Suggested change has been incorporated
Comment 4 -Please add the aim of the manuscript in the introduction before starting 1.1.
Response- Suggested change has been incorporated (Line 86-96 )
Comment 5 -In 1.2.2, either please add few more studies or elaborate the study of Li and Handler 2019 in detail. Same comment is applicable for 1.2.3; 1.3.3; 1.3.4; 1.3.5; 1.3.6; 1.5.2. Please describe the results of the study in 1.4.3.
Response- Suggested change has been incorporated (line-217-245, line 338-378, line 495-505, line 523-531)
Comment 6 -Please describe some information on 1.6. and 1.7 as it was provided for 1.2, 1.3, 1.4 etc. to maintain the symmetry.
Response- Suggested change has been incorporated (line- 425-428 and 446-447)
Comment7 -Please modify the Line 422.
Response- Suggested change has been incorporated
Comment 8 -Before conclusions, it will be better to add a separate heading describing the challenges in the application of CRISPR technique for insect pest management.
Response- Suggested change has been incorporated (line 469-490)
Please see the attachment.

Reviewer 2 Report
The authors highlighted the threat of pests to agriculture. Furthermore, CRISPR technology reduces the pests' resistance to insecticides, compromises the pest's fitness to reproduce, and hinders the pest's metamorphosis. There are several questions that arise while reading this manuscript:
1. Line 176-177 “Further, the information from this study was exploited for controlling three more insect pests of the same family Anastrepha, Bactrocera, and Ceratitis capitata”. Please describe briefly the information used for controlling the insect pest
2. The review often resembles the research article "Progress and Prospects of CRISPR/Cas Systems in Insects and Other Arthropods". Many of the headings and content appear to be adapted from the existing review. It looks like only paraphrasing is done for information on Tribolium castaneum and Dendrolimus punctatus. Please comment on this.
3. Already there are many reviews available on CRISPR/Cas and insect/pest management. How this manuscript is different and updated will attract readers.
Author Response
Replies To Comments By Reviewer 2
We are grateful to the reviewer for their critical evaluation and valuable suggestions. All the suggested changes have been incorporated. We believe that the Ms now meets to your expectation.
Comment 1. Line 176-177 “Further, the information from this study was exploited for controlling three more insect pests of the same family Anastrepha, Bactrocera, and Ceratitis capitata”. Please describe briefly the information used for controlling the insect pest-
Response- Suggested change has been incorporated (line 212 to216).
Comment 2. The review often resembles the research article "Progress and Prospects of CRISPR/Cas Systems in Insects and Other Arthropods". Many of the headings and content appear to be adapted from the existing review. It looks like only paraphrasing is done for information on Tribolium castaneum and Dendrolimus punctatus. Please comment on this.
Response-
- Our MS strictly highlights the studies that have applied CRISPR/Cas9 for insect pest control. The above-mentioned review on the other hand has emphasised on gene function exploration and experimental optimization and few applied studies.
- The above-mentioned paper has reviewed studies done on non-pest insects also whereas our MS is restricted to pests only.
- Our MS does not talk about experiment optimization and off target effects because there is already a lot of literature available on these two topics.
- In the MS we have even proposed two potential ideas (line 584-603) for future research work in the field.
- For Tribolium castaneum and Dendrolimus punctatus we realize the information is overlapping but, our MS have reviewed only the pest control studies and not the ones which did the functional gene analysis. Also, the data in our MS is updated.
- The present review article will help the researcher to get a comprehensive outlook of this technology to be utilized for successful control of agricultural pests and sustainable development.
Comment 3. Already there are many reviews available on CRISPR/Cas and insect/pest management. How this manuscript is different and updated will attract readers.
Response-
- As per best to our knowledge no research group has covered such a wide range of insect pests as our MS has done.
- The points mentioned in “Comment No. 2” also answer the “Comment No. 3”.
Please see the attachment

Reviewer 3 Report
The article has worth publishing.
But here are a few comments,
The use of the word "stolen" at line no. 81 is not appropriate. such inappropriate comments please be carefully checked throughout the manuscript.
Somewhere spacing is an issue between the sentences and insertion of a full stop at the wrong place.
Species names should also be italized in section headings.
The conclusion with recommendations should be added.
Figures visualization needs slight improvement (optional)
After these corrections paper is recommended for publication.

Author Response
Replies To Comments By Reviewer 3
We are grateful to the reviewer for their critical evaluation and valuable suggestions. All the suggested changes have been incorporated. We believe that the Ms now meets to your expectation.
Comment 1- The use of the word "stolen" at line no. 81 is not appropriate. such inappropriate comments please be carefully checked throughout the manuscript.
Response- Suggestion incorporated (line 107).
Comment 2- Somewhere spacing is an issue between the sentences and insertion of a full stop at the wrong place.
Response- Suggestion incorporated.
Comment 3- Species names should also be italized in section headings.
Response- Suggestion incorporated.
Comment 4- The conclusion with recommendations should be added.
Response- Suggestion incorporated (line- 587-624).
Comment 5- Figures visualization needs slight improvement (optional)
Response- Suggestion incorporated.
Comment 6- What is the main question addressed by the research?
The articles is titled “CRISPR/Cas9 for insect pests management: a comprehensive review of advances and applications”. This work has the main focus on the power of CRISPR/Cas9 and proposes potential research ideas for CRISPR/Cas9-based integrated pest management. Thus, the objective of the review seems quite broad and seem author have tried to compile the literature cited where CRISPR has been tested in insets which are the pests. Better to define a few aspects like insecticide resistance, may be highlighted in the objective to make more specific and understand the scope of the review. They are suggested either to modify the title or modify the objectives to restructure the article flow. Authors have rarely discussed the main concept of the technique, its advances (e.g. CRISPR editor), advantages and limitations.
Responses- The title and central idea of the work is distinct from the previously available reviews as our MS is mainly and strictly focused on the application of CRISPR/Cas9 technology for insect pest management. Present review article is modified and updated according to the suggestions to suit the title of the review.
We have avoided discussing the concepts, advantages and limitations of the CRISPR/Cas9 technology as, a lot of literature is available on these topics.
Comment 7-. Do you consider the topic original or relevant in the field? Does it address a specific gap in the field?
No. The theme of the topic does not stand distinct from published review. Topics does not justify the gap. Why is need this paper is not properly satisfied. It has been noticed below papers are published and comprehensively addressed the essence of this reviewer paper. It is beyond understanding why have they missed to cite these important reviewers. They must compared the earlier review to scope of current review and fully justify the gap which is seriously missing.
Responses-
- As per best of our knowledge no research group has covered such a wide range of insect pests as present MS has done.
- Our MS strictly highlights the studies that have applied CRISPR/Cas9 for insect pest control. The previous reviews on the other hand emphasized on gene function exploration and experimental optimization and few applied studies.
- The previous reviews have reviewed studies done on non-pest insects also whereas our MS is restricted to pests only.
- Our MS does not talk about experiment optimization and off target effects because there is already a lot of literature available on these two topics.
- In the MS we have even proposed two potential ideas (line 584-603) for future research work in the field.
Comment 8-. What does it add to the subject area compared with other published material?
Comparing with the published review, it’s very hard to distinguish their contribution. As suggested in point 2, only after additional argument by author we have to understand the contribution.
This work simply have tried to list down the literature but have not addressed the critical challenge for implementing the CRISPR/Cas9 in pest, least is mentioned on the off target as one of the limitation of the CRISPR limitation. The ethical issues concerned have least stated in paper which need more attention.
Response
- Our MS have reviewed only the pest control studies and not the ones which did the functional gene analysis. Also, the data in our MS is updated.
- The present review article will help the researcher to get a comprehensive outlook of this technology to be utilized for successful control of agricultural pests and sustainable development.
Li, Jiang-Jie, et al. "CRISPR/Cas9 in lepidopteran insects: Progress, application and prospects." Journal of Insect Physiology 135 (2021): 104325. Not cited –
Response-suggestion incorporated
Douris, Vassilis, et al. "Using CRISPR/Cas9 genome modification to understand the genetic basis of insecticide resistance: Drosophila and beyond." Pesticide biochemistry and physiology 167 (2020): 104595. Not cited-
Response- suggestion incorporated
Sun, Dan, et al. "Progress and prospects of CRISPR/Cas systems in insects and other arthropods." Frontiers in physiology 8 (2017): 608. Not cited –
Response- Please check reference 2
Gantz, Valentino M., and Omar S. Akbari. "Gene editing technologies and applications for insects." Current opinion in insect science 28 (2018): 66-72.
Response- suggestion incorporated
Comment 9-. What specific improvements should the authors consider regarding the methodology? What further controls should be considered?
They have to provide chronological development graph clearly indicate breakthrough in CRISPR use in pest management. Although, it is a review article the structure of presentation is different. But author is suggested to address appropriately what are those heading or area which this review is to focus. Discussing separately each species or class of insect do not provide an integrated and comprehensive approach.
Response- For every insect we have maintained chronological development in both the table and the text. We believe making a chronological development graph will be a simple repetition of the information. Still, if the reviewer finds it essential we can add the suggestion.
Comment 10-. Are the conclusions consistent with the evidence and arguments presented and do they address the main question posed?
Further on page 9 the reference and figures are cited in reference which is wrong.
Response- Suggestion incorporated.
Page 9 and 17 have two heading for conclusion. Conclusion is too generic.
Response- the “conclusion” heading on the 17th page was a part of the format provided by the journal. It has been removed. We have further modified and changed the conclusion to discussion.
Key outcome are need to readdress with more of detail of strength and limits and way forward with some proposal for scientific community to which direction of the research should go.
Response- In the present MS we have even proposed two potential ideas (line 584-603) for future research work in the field.
Comment 11-. Are the references appropriate?
Have to update with latest 2022 references. Seems biased with selection of references. As mentioned above many of review on same topic are not cited and explain what is new in their proposal. –
Response- Suggestion incorporated, new references added.
Please include any additional comments on the tables and figures.
Comment 12- In Table 1, the gene accession number should be added
Response- We have added the accession number of the target genes since not all papers have cited the accession numbers. Also, some of the sequences were obtained from unpublished data, therefore they could not be added in the table.
Comment 13 - the yellow name of the gene or the color.
Response- The yellow gene is named after its effect on pigmentation on mutation. Its mutation reduces male mating success.
Comment 14- The compressive design of how CRIPSR is implemented is missing and what are the modified approaches of CRISPR used mechanism is lacking. Hence, need to draw an integrated or two separate figures.
Response- In figure 2 authors have made an effort to represent the implementation of CRISPR/Cas9 system. We understand that molecular level is not represented but a detailed pictorial representation of implementation and consequences are shown. The modified approaches of CRISPR used mechanism are discussed in the column “Delivery of CRISPR components” in the table1.
Please see the attachment

Round 2
Reviewer 1 Report
The authors have incorporated the suggestions.